# Regulatory Networks of Flowering Genes in *Angelica sinensis* during Vernalization

**DOI:** 10.3390/plants11101355

**Published:** 2022-05-19

**Authors:** Mimi Luo, Xiaoxia Liu, Hongyan Su, Meiling Li, Mengfei Li, Jianhe Wei

**Affiliations:** 1State Key Laboratory of Aridland Crop Science, College of Life Science and Technology, Gansu Agricultural University, Lanzhou 730070, China; luomimi9521@163.com (M.L.); 18893481962@163.com (X.L.); shy922322@163.com (H.S.); mlli1996@163.com (M.L.); 2Institute of Medicinal Plant Development, Chinese Academy of Medical Sciences & Peking Union Medical College, Beijing 100193, China

**Keywords:** *Angelica sinensis*, vernalization, flowering, transcriptomic, gene-regulatory networks

## Abstract

*Angelica sinensis* is a low-temperature and long-day perennial herb that has been widely used for cardio-cerebrovascular diseases in recent years. In commercial cultivation, up to 40% of flowering decreases the officinal yield of roots and accumulation of bioactive compounds. Although the regulatory mechanism of flowering genes during the photoperiod has been revealed, the networks during vernalization have not been mapped. Here, transcriptomics profiles of *A. sinensis* with uncompleted (T1), completed (T2) and avoided vernalization (T3) were performed using RNA-seq, and genes expression was validated with qRT-PCR. A total of 61,241 isoforms were annotated on KEGG, KOG, Nr and Swiss-Prot databases; 4212 and 5301 differentially expressed genes (DEGs) were observed; and 151 and 155 genes involved in flowering were dug out at T2 vs. T1 and T3 vs. T1, respectively. According to functional annotation, 104 co-expressed genes were classified into six categories: FLC expression (22; e.g., *VILs*, *FCA* and *FLK*), sucrose metabolism (12; e.g., *TPSs*, *SUS3* and *SPSs*), hormone response (18; e.g., *GID1B*, *RAP2s* and *IAAs*), circadian clock (2; i.e., *ELF3* and *COR27*), downstream floral integrators and meristem identity (15; e.g., *SOC1*, *AGL65* and *SPLs*) and cold response (35; e.g., *PYLs*, *ERFs* and *CORs*). The expression levels of candidate genes were almost consistent with FPKM values and changes in sugar and hormone contents. Based on their functions, four pathways that regulate flowering during vernalization were mapped, including the vernalization pathway, the autonomic pathway, the age pathway and the GA (hormone) pathway. This transcriptomic analysis provides new insights into the gene-regulatory networks of flowering in *A. sinensis*.

## 1. Introduction

*Angelica sinensis* (Oliv.) Diels (Apiaceae, alt. Umbelliferae) is a perennial herbaceous species, originally native to China, with a dense population in Gansu province at an altitude of 2200–3000 m [1]. The roots have been widely used as a traditional Chinese herbal remedy in China, Korea and Japan for nourishing the blood, regulating menstrual disorders, relaxing bowels, etc., for over 2000 years [2,3]. Currently, over 50 compounds have been isolated from the roots, including phthalides, organic acids and polysaccharides, which confer pharmacological effects, including cardio-cerebrovascular, hepatoprotective, antioxidant, antispasmodic and immunomodulatory effects [4,5,6].

At present, over 43,500 ha of commercial of *A. sinensis* cultivation supplies the increasing market demand; however, up to 40% of flowering of plants in the field makes the roots lignified, decreasing the yield of roots and the accumulation bioactive compounds [7,8,9]. Previous studies have demonstrated that *A. sinensis* is a low-temperature and long-day plant that must experience vernalization (0 to 5 °C) and long-day conditions (>12 h daylight) for the transition from vegetative growth to flowering [10]. Thereafter, the flowering could be effectively avoided after the seedlings stored below freezing temperature (<0 °C) [11,12,13] or significantly inhibited when the plants grew under sunshade at photoperiod [14].

To date, the gene regulatory mechanism of the flowering during the photoperiod has been shown, with upstream regulatory pathway genes (e.g., *CONSTANS* (*CO*), *FLOWERING LOCUS C* (*FLC*) and *PHYOCHROME A* (*PHYA*)), downstream floral integrator genes (e.g., *FLOWERING LOCUS T* (*FT*), *SUPPRESSOR OF OVEREXPRESSION OF CONSTANS 1* (*SOC1*) and *FT-interacting protein 1* (*FTIP1*)), downstream floral meristem identity genes (e.g., *LEAFY* (*LFY*), *APETALA 1* (*AP1*) and *AP2*)), autonomous pathway genes (e.g., *flowering time control protein FCA* (*FCA*), *flowering*
*time*
*control*
*protein FPA* (*FPA*) and *FY*), sucrose metabolism genes (e.g., *sucrose synthase* (*SUS*), *alpha-amylase* (*AMY*) and *glucan endo-1,3-b-glucosidase* (*GLU*)) and gibberellin (GA) pathway genes (e.g., *gibberellin 2-β-dioxygenase 1* (*GA2OX1*), *gibberellin 20-oxidase 1* (*GA20OX1*) and *DELLA protein GAI* (*GAI*)) [15,16,17,18]. Although physiological changes (e.g., soluble sugar, protein and hormones) during vernalization and freezing storage have been investigated [19,20], gene-regulatory networks involved in the flowering of *A. sinensis* during vernalization have not been mapped.

In the present study, the alterations of transcripts in *A. sinensis* with three different treatments, including uncompleted (T1), completed (T2) and avoided vernalization (T3) were analyzed by combining isoforms with Illumina sequencing, and 104 DEGs involved in flowering and cold response were observed. The changes in soluble sugar and hormones (GA_3_, IAA and ABA) were examined via a spectrophotometer and HPLC.

## 2. Materials and Methods

### 2.1. Plant Material

The seedlings (root-tip size 0.4–−0.5 cm; Appendix A) of *Angelica sinensis* (cultivar Mingui 1) were, respectively, stored at 0 and −3 °C on 21 October 2020. The seedlings stored at 0 °C were collected after 14 (T1) and 60 days (T2), and the seedlings stored at −3 °C were collected after 125 days (T3). The shoot apical meristem (SAM) of the collected seedlings was immediately frozen and stored in liquid nitrogen for transcriptomic analysis and qRT-PCR validation. Herein, T1 (0 °C 14 d), T2 (0 °C 60 d) and T3 (−3 °C 125 d) treatments represent uncompleted, completed and avoided vernalization, respectively (Appendix A).

### 2.2. Isoform Sequencing and Analysis

The total RNA of SAM was extracted using a Trizol reagent, the integrity of the RNA was determined using an Agilent 2100 Bioanalyzer and agarose gel electrophoresis, and the purity and concentration of the RNA were determined using a NanoDrop micro-spectrophotometer. mRNA was enriched by Oligo (dT) magnetic beads and then reverse-transcribed into cDNA using a Clontech SMARTer cDNA Synthesis Kit. The cDNA was amplified by PCR for 13 cycles to generate large-scale, double-stranded cDNA and then purified using AMPure XP beads to select cDNAs > 4 kb in size. SMRTbell library was constructed after the cDNAs were damage-repaired and end-repaired, and the sequencing adapters were ligated. The SMRTbell template was annealed to bind the primer and polymerase.

The raw reads of the cDNA library were sequenced on a Pacific Biosciences Sequel platform by Gene Denovo Biotechnology Co., Ltd. (Guangzhou, China), and then the raw reads were analyzed using a SMRT Link (V8.0.0) [21]. Briefly, high-quality circular consensus sequences (CCS) were extracted from the raw reads, and full-length non-chimeric (FLNC) reads were obtained by removing the primers, barcodes, poly (A) tail trimmings and concatemers. Then, the FLNC reads were clustered to generate the entire isoforms [22]. Similar FLNC reads were used, the minimap2 was used to cluster hierarchically to obtain the consistency sequence (unpolished consensus isoforms), the quiver algorithm was used to further correct the consistency of the sequence and then high-quality isoforms were obtained with a prediction accuracy ≥0.99. Finally, the full-length isoforms were analyzed and annotated against the databases using BLAST: NCBI non-redundant protein (Nr) (http://www.ncbi.nlm.nih.gov, accessed on 1 December 2021), Kyoto Encyclopedia of Genes and Genomes (KEGG) (http://www.genome.jp/kegg, accessed on 1 December 2021), Eukaryotic Orthologous Groups of proteins (KOG), Swiss-Prot (https://www.uniprot.org, accessed on 1 December 2021), Gene Ontology (GO) (http://www.geneontology.org/, accessed on 1 December 2021) and COG/KOG (http://www.ncbi.nlm.nih.gov/COG, accessed on 1 December 2021) with the BLASTx program (http://www.ncbi.nlm.nih.gov/BLAST/, accessed on 1 December 2021) at an E-value threshold of 1 × 10^−5^ to evaluate the sequence similarity with genes of other species.

### 2.3. Illumina Sequencing and DEGs Analysis

The procedures of total RNA extraction, integrity and purity determination, as well as cDNA library construction, for the samples T1, T2 and T3 were conducted according to the “Section 2.2” description. After the samples were sequenced, the raw reads were filtered using a FASTQ system to obtain high-quality clean reads [23] with the following parameters: removing reads containing adapters, removing reads containing more than 10% unknown nucleotides (N), and removing low-quality reads containing more than 50% low quality (Q-value ≤ 20) bases. The paired-end clean reads were mapped to the full-length isoforms using HISAT2. 2.4 to obtain the unique- and multiple-mapped reads [24] with “-rna-strandness RF” and other parameters set as a default. 

The expression level of each transcript was normalized to the values of the fragments per kilobase of the exon model per million mapped reads (FPKM). Differential expression analysis of transcripts was performed using DESeq2 software [25] between different groups. The differential expression levels at T2 vs. T1 and T3 vs. T1 were determined with the criteria of the false discovery rate (FDR) < 0.05 and |log_2_ (fold-change)| > 1. To date, the genome of *A. sinensis* has not been reported to be sequenced and assembled. The function of DEGs was analyzed using BLAST against the full-length isoforms of *A. sinensis* based on the Nr, KEGG, KOG and Swiss-Prot databases.

For GO enrichment analysis, all DEGs were mapped to GO terms in the GO database, gene numbers were calculated (FDR ≤ 0.05) for every term, and GO terms that were significantly enriched in DEGs compared with the full-length isoforms of *A. sinensis* were defined using a hypergeometric test. For the KEGG pathway enrichment analysis, significantly enriched metabolic pathways or signal transduction pathways in DEGs compared with the full-length isoforms of *A. sinensis* were defined with FDR ≤ 0.05 as a threshold.

### 2.4. qRT-PCR Validation

The procedures of total RNA extraction, integrity and purity determination for the samples T1, T2 and T3 were conducted according to the description labeled “Section 2.2”. Primer sequences of selected DEGs (Appendix A) were designed using an NCBI primer-blast tool and synthesized by Sangon Biotech (Shanghai, China). First-strand cDNA was synthesized using a FastKing RT kit (KR116; Tiangen, Beijing, China) with one cycle at 42 °C for 15 min and then at 95 °C for 3 min. qRT-PCR gene expression was carried out using a SuperReal PreMix Plus (SYBR Green) (FP205; Tiangen, Beijing, China) with one cycle at 95 °C for 15 min, followed by 40 cycles at 95 °C for 10 s, 60 °C for 20 s and 72 °C for 30 s. Melting curve analysis was performed after incubation at 95 °C for 15 s, 60 °C for 1 min and 95 °C for 1 s. The *actin* gene was used as a reference control gene [9,15,16,26]. Herein, the cycle threshold (Ct) values and standard curves of the *ACT* gene at different volumes (0.25, 0.5, 1.0, 1.5, 2.0 and 3.0 μL) were built to correct the gene expression level (Appendix A). The relative expression level (REL) of genes was calculated using a 2^−ΔΔCt^ method (Ct, cycle threshold value of target gene) [27].

### 2.5. Measurement of Soluble Sugar and Endogenous Hormones Contents

Soluble sugar content was measured using phenol-sulfuric acid [28]. Endogenous hormones (GA_3_, IAA and ABA) contents were measured using a HPLC method [29].

### 2.6. Statistical Analysis

In order to obtain the precise estimation of PCR efficiency, each experiment for qRT-PCR validation was performed with three biological replicates, along with three technical replicates. Statistical analysis was performed via ANOVA and Duncan multiple comparison tests, and SPSS 22.0 was the software package used with *p* < 0.05 as the basis for statistical differences.

## 3. Results

### 3.1. Full-Length Isoform Analysis

A total of 1,031,219 high-fidelity reads were extracted after 33 full passes of raw reads, 76,840 polished high-quality isoforms were obtained using a Quiver calculation and 64,117 full-length isoforms were generated after the FLNC reads were clustered (Appendix A). A total of 61,241 isoforms were annotated on KEGG (60,792), KOG (42,645), Nr (61,161) and Swiss-Prot (51,070) databases (Figure 1A), and the top 10 species in terms of distribution against Nr, were *Daucus carota*, *Apium graveolens*, *Nyssa sinensis*, *Angelica sinensis*, *Petroselinum crispum*, *Camellia sinensis*, *Mikania micrantha*, *Vitis vinifera*, *Oliveria decumbens* and *Beta vulgaris* (Figure 1B). Since the genome of *A. sinensis* has not been sequenced, the isoforms are needed to compare with other species and the limited genes of *A. sinensis* in the NR database.

### 3.2. Illumina Sequencing of T1, T2 and T3

To reveal the molecular mechanisms responsible for regulating the flowering of *A. sinensis* during vernalization, a comparison of gene transcription in response to different temperatures and durations was performed. In this study, 9.14, 9.19 and 8.63 million raw reads were generated for T1, T2 and T3, respectively. After raw data filtering, 9.12, 9.16 and 8.61 million clean reads were collected. After mapping on the isoforms, 5.33, 5.50 and 5.23 million unique mapped reads, as well as 2.27, 2.31 and 2.05 million multiple mapped reads, were obtained from the T1, T2 and T3, respectively. Meanwhile, the exon rate of all the three treatments reached 100% (Table 1).

### 3.3. Analysis and Annotation of Differentially Expressed Genes (DEGs)

#### 3.3.1. DEGs at T2 vs. T1 and T3 vs. T1

A total of 4212 and 5301 DEGs were observed from the 61,241 isoforms, with 1701 up-regulated (UR) and 2511 down-regulated (DR) at T2 vs. T1, and 2544 UR and 2757 DR at T3 vs. T1 (Figure 2), based on the FPKM values (Appendix A), principal component analysis (PCA) (Appendix A) and Pearson correlation analysis (Appendix A). The cluster heat map of the DEGs at T2 vs. T1 and T3 vs. T1 was shown in Figure 3, which provides further analysis such as identification of the gene functions or gene response analysis.

#### 3.3.2. GO and KEGG Enrichments of DEGs

Based on the GO system, the DEGs were classified into three ontologies including biological process (BP), cellular component (CC) and molecular function (MF) (Appendix A). Based on the KEGG database, at T2 vs. T1, the 4212 DEGs were enriched into 127 pathways, such as plant hormone signal transduction, starch and sucrose metabolism, and phenylpropanoid biosynthesis; at T3 vs. T1, the 5301 DEGs were enriched into 128 pathways, such as biosynthesis of secondary metabolites, the MAPK signaling pathway, and plant and starch and sucrose metabolism (Appendix A).

#### 3.3.3. Uncovering DEGs Involved in Flowering during Vernalization

Based on the regulatory pathways of flowering genes in *Arabidopsis* [30,31,32], 151 and 155 genes involved in regulating flowering were, respectively, uncovered from the 4212 DEGs at T2 vs. T1 and 5301 DEGs at T3 vs. T1, and 104 co-expressed genes were classified into six categories: FLC expression (22), sucrose metabolism (12), hormone response (18), circadian clock (2), downstream floral integrators and meristem identity (15), and cold response (35) (Figure 4). The 104 DEGs exhibited a −3.68- to 4.07-fold and −8.75- to 9.48-fold differential expression at T2 vs. T1 and T3 vs. T1, respectively (Figure 5). The sequence details of the isoforms involved in the 104 co-expressed genes are shown in Appendix A.

#### 3.3.4. DEGs Involved in FLC Expression

Out of the 104 DEGs involved in regulatory pathways of flowering, 22 genes were involved in FLC expression (Figure 5A, Appendix A), with 11 genes inhibiting FLC expression, including vernalization response (*VIL1*, *VIL2* and *VRN1-L*), autonomous pathway [RNA binding (*FCA*, *FLK* and *FPA*); RNA process (*FY*) and chromatin modification (*FLD* and *LD*)], chromo domain protein (*LHP1*) and repressor of RNA polymerase III transcription (*MAF1*); 9 genes activating FLC expression, including PAF1 complex (*VIP4*, *ATX2*, *ATX4* and *ATX5*), SWR1 complex (*ARP6* and *PIE1*) and FRIGIDA-like protein (*FRL3*, *FES1* and *SUF4*); and 2 genes interacting with FLC expression, including polycomb group (PcG) protein (*CLF* and *FIE2*). Fifteen genes were confirmed by qRT-PCR: the RELs of 11 genes inhibiting FLC expression (*VIL1*, *VIL2*, *VRN1-L*, *FCA*, *FLK*, *FPA*, *FY*, *FLD*, *LD*, *LHP1* and *MAF1*) and 4 genes activating FLC expression (*VIP4*, *ATX2*, *ARP6* and *FRL3*) were almost consistent with their FPKM values at T2 vs. T1 and T3 vs. T1 (Figure 5A and Figure 6).

#### 3.3.5. Genes Involved in Sucrose Metabolism

Twelve genes involved in sucrose metabolism were observed to be differentially expressed (Figure 5B, Appendix A), with genes encoding trehalose-phosphate synthase (*TPS1*, *TPS5*, *TPS7* and *TPS10*), sucrose synthase (*SUS2* and *SUS3*), sucrose-phosphate synthase (*SPS* and *SPS1*), phosphoglucomutase (*PGMP* and *PGM1*) and beta-amylase (*BAM1* and *BAM3*). Three genes were confirmed using qRT-PCR: the RELs of these three genes (*TPS1*, *SUS3* and *SPS*) were consistent with their FPKM values at T2 vs. T1 and T3 vs. T1 (Figure 5B and Figure 7A). The soluble sugar content exhibited a 0.77-fold decrease at T2 vs. T1 and a 1.13-fold increase at T3 vs. T1 (Figure 7B).

#### 3.3.6. Genes Involved in Hormone Response

Eighteen genes involved in hormone response were observed to be differentially expressed (Figure 5C, Appendix A), with gibberellin (GA) response (*GID1B*), ethylene (ETH) response (*RAP2-3* and *RAP2-7*), ndole-3-acetic acid (IAA)-amido synthetase (*GH3.6*), auxin transport (*LAX2*), auxin response (*IAA8*, *IAA9*, *IAA12*, *IAA13*, *IAA27*, *ARF1*, *ARF4*, *ARF5*, *ARF6* and *ARF19*) and cytokinin (CTK) response (*AHK3*, *AHP1* and *ARR2*). Five genes were confirmed by qRT-PCR, and the RELs of these five genes (*GID1B*, *RAP2-7*, *IAA13*, *ARF1* and *AHK3*) were consistent with their FPKM values at T2 vs. T1 and T3 vs. T1 (Figure 5C and Figure 8A). The contents of GA_3_ and IAA exhibited a 1.86- and 1.59-fold increase at T2 vs. T1 and a 0.42- and 0.65-fold decrease at T3 vs. T1, respectively (Figure 8B,C).

#### 3.3.7. DEGs Involved in Circadian Clock

Two genes involved in circadian clock were observed to be differentially expressed (Figure 5D, Appendix A), including a transcription factor part of a circadian clock (*ELF3*) and cold-regulated protein (*COR27*). These two genes were confirmed using qRT-PCR, and the RELs were consistent with their FPKM values at T2 vs. T1 and T3 vs. T1 (Figure 5D and Figure 9A).

#### 3.3.8. DEGs Involved in Downstream Floral Integrators and Meristem Identity

Fifteen genes involved in downstream floral integrators and meristem identity were observed to be differentially expressed (Figure 5E, Appendix A); these included floral integrators (*FTIP3*, *FTIP7* and *SOC1*) and floral meristem identity (*AGL65*, *SPL1*, *SPL6*, *AP2*, *AP2-1*, *AP2-2*, *AP2-3*, *AIL5*, *ANT*, *At2g41710*, *TEM1* and *SOK2*). Six genes were confirmed by qRT-PCR, and the RELs of six of these genes (*FTIP3*, *FTIP7*, *SOC1*, *AGL65*, *SPL1* and *SOK2*) were consistent with their FPKM values at T2 vs. T1 and T3 vs. T1 (Figure 5E and Figure 9B).

#### 3.3.9. DEGs Involved in Cold Response

Thirty-five genes involved in cold-stress response were observed to be differentially expressed (Figure 5F, Appendix A), with abscisic acid (ABA) response (*PYL3* and *PYL4*), ETH response (*ERF3*, *ERF5*, *ERF010*, *ERF011*, *ERF13*, *ERF110*, *ERF118* and *ETR2*), cold-regulated plasma membrane proteins (*COR413PM1* and *COR413PM2*), cold-responsive protein kinase (*CRPK1*, *MPK3*, *MPK9*, *MPK16*, *YDA* and *SRK2E*), protein phosphatase (*PP2CA*, *HAB1*, *HAB2*, *KAPP*, *PPC6-1*, *PPC6-7*, *PP2C27*, *PP2C38*, *At2g30020*, *Os02g0799000*, *At3g15260*, *At3g16560*, *At3g62260*, *Os06g0651600* and *At4g31860*), and other stress-response factors and protein (*TGA2.2* and *CDC5*). Nine genes were confirmed by qRT-PCR, and the RELs of these nine genes (*PYL3*, *ERF110*, *ETR2*, *COR413PM1*, *CRPK1*, *MPK3*, *SRK2E*, *PP2CA* and *HAB1*) were almost consistent with their FPKM values at T2 vs. T1 and T3 vs. T1 (Figure 5F and Figure 10A). The ABA content exhibited a 0.49- and 0.84-fold decrease at T2 vs. T1 and T3 vs. T1, respectively (Figure 10B).

## 4. Discussion

Vernalization is the process whereby flowering is promoted by prolonged exposure to a cold treatment given to a fully hydrated seed or a growing plant; without vernalization, plants show delayed flowering or remain vegetative. Meanwhile, vernalization can be lost at high temperatures and avoided below freezing temperatures [33]. Extensive studies have reported that vernalization suppresses the expression of genes encoding the repressors of flowering [34]; moreover, the gene network of flowering time has been mapped with known genetic and epigenetic regulators in the model plant *Arabidopsis thaliana* [31,32]. For *A. sinensis*, the effective temperature range for vernalization is from 0 to 5 °C with a duration 57 to 85 d, and the vernalization can be effectively avoided with exposure of seedlings to temperatures from −2 to −12 °C [10,11,12,13]. However, gene-regulatory networks involved in the flowering of *A. sinensis* during the vernalization have not been revealed. By storing seedlings at 0 °C for 14 d (uncompleted vernalization, T1) and 60 d (completed vernalization, T2), as well as at −3 °C for 125 d (avoided vernalization, T3), a total of 4212 and 5301 genes were differentially expressed, and 151 and 155 genes were involved in flowering at T2 vs. T1 and T3 vs. T1, respectively. Based on their biological functions, 104 co-expressed genes were classified into 6 categories: FLC expression, sucrose metabolism, hormone response, circadian clock, downstream floral integrators and meristem identity, and cold response.

### 4.1. Genes Involved in FLC Expression

FLC is a MADS-box transcriptional regulator that acts as a potent repressor of flowering [35]. Several genes inhibit FLC expression; among these, 11 genes encoding proteins are involved in vernalization response, including VIN3-like proteins (VIL1 and VIL2) that inhibit the expression of FLC and FLM, which are associated with an epigenetically silenced state and with acquisition of competence to flower [36,37,38,39], and VRN1-like (VRN1-L), which may act as transcriptional repressor of FLC [40]. The genes that inhibit FLC expression also include the genes involved in the autonomous pathway, including proteins that control flowering time, FCA and FPA, which decrease FLC expression and act redundantly with each other to prevent the expression of distally polyadenylated antisense RNAs at the FLC locus [41,42,43]; flowering locus K (FLK), which represses FLC expression and post-transcriptional modification [44,45]; another protein controlling flowering time, FY, which decreases FLC expression and is required for the negative autoregulation of FCA expression [46,47]; and FLOWERING LOCUS D (FLD) and LUMINIDEPENDENS (LD), which decrease FLC expression via chromatin modification [48,49,50]. Finally, there are two other genes that inhibit FLC expression: chromo domain protein LHP1 (LHP1), which is a structural component of heterochromatin involved in gene repression via methylating H3-K9 [51], and MAF1 homolog (MAF1), which is an element of the TORC10 signaling pathway and represses RNA polymerase III transcription [52].

For genes activating FLC expression, there are nine genes encoding proteins involved in the PAF1 complex: LEO1 homolog (VIP4), which is involved in histone modifications and is required for the FLC expression [53,54]; histone-lysine N-methyltransferases (ATX2, ATX4 and ATX5), which dimethylate H3K4me2 and are involved in epigenetic regulation of FLC and FT [55,56,57]; genes involved in the SWR1 complex, namely actin-related protein 6 (ARP6) and PHOTOPERIOD-INDEPENDENT EARLY FLOWERING 1 (PIE1), which are associated with transcriptional regulation of selected genes (e.g., *FLC*) via chromatin remodeling; ARP6, which is required for the activation of *FLC* and *FLC/MAF* genes expression through both histone H3 and H4 acetylation and methylation [58,59]; PIE1, which is required for the reactivation of *FLC* [60,61]. There are also genes involved in the FRIGIDA-like protein: FRIGIDA-like protein 3 (FRL3), which trimethylates H3K4, increasing *FLC* expression [62]; FRIGIDA-ESSENTIAL 1 (FES1), which acts cooperatively with *FRI* or *FRL1* to promote *FLC* expression [63,64]; and SUPPRESSOR OF FRI 4 (SUF4), which recruits the FRI-C complex to the FLC promoter and is required for FRI-mediated FLC activation and maintains high levels of *FLC* expression [63,65]. 

There are also genes interacting with FLC expression: these include two genes involved in PcG protein: histone-lysine N-methyltransferase CLF (CLF) and PcG protein FIE1 (FIE2), which are required to maintain the transcriptionally repressive state of homeotic genes by methylating H3-K27, rendering chromatin heritably changed in its expressability [66,67]. According to the RELs of genes calculated with the FPKM value and validated using qRT-PCR, the genes involved in inhibiting *FLC* expression were up-regulated, while the genes involved in activating *FLC* expression were down-regulated at T2 vs. T1 (during vernalization); however, these genes showed the opposite expression levels at T3 vs. T1 (freezing temperature avoided vernalization) (Figure 6; Appendix A). These findings are consistent with previous studies with higher flowering rates at T2 with lower flowering rates at T3 compared with T1 (Appendix A).

### 4.2. Genes Involved in Sucrose Metabolism

For vernalization to occur, sources of energy (sugars) and carbohydrate metabolism are required [33,68]. Twelve genes were found to be involved in sucrose metabolism: trehalose-phosphate synthases (TPS1, TPS5, TPS7 and TPS10), which are required for vegetative growth and transition to flowering by regulating starch and sucrose degradation [69,70,71]; sucrose synthases (SUS2 and SUS3), which provide UDP glucose and fructose for various metabolic pathways [72]; sucrose-phosphate synthases (SPS and SPS1), which regulate sucrose synthesis from UDP glucose and fructose-6-phosphate [73]; phosphoglucomutases (PGMP and PGM1), which participate in both the breakdown and the synthesis of glucose [74]; and beta-amylases (BAM1 and BAM3), which play roles in circadian-regulated starch degradation and maltose metabolism [75]. According to the RELs, the genes involved in sucrose and starch degradation were up-regulated during vernalization (T2) and down-regulated at freezing temperature (T3) compared with T1, while the genes involved in sucrose and starch biosynthesis showed the opposite expression levels (Figure 7A; Appendix A). This differential expression of genes is consistent with the change in soluble sugar contents at T1, T2 and T3 (Figure 7B).

### 4.3. Genes Involved in Hormone Response

The GA pathway is required for early flowering by promoting the expression of the *LFY* gene; meanwhile, other growth hormones (e.g., ETH, IAA and CTK) can either inhibit or promote flowering [33]. Here, there were 18 genes that were found to be involved in the hormone response, including the GA receptor GID1B (GID1B), which functions as soluble GA receptor interacting with specific DELLA proteins, known as repressors of GA-induced growth and flower development [76]; ETH-responsive transcription factors (RAP2-3 and RAP2-7), which negatively regulate the transition to flowering time and cause a delay in flowering time [77]; IAA-amido synthetase (GH3.6), which is involved in auxin signal transduction [78]; auxin transporter-like protein 2 (LAX2), which is involved in proton-driven auxin influx and basipetal auxin transport [79,80]; Aux/IAA proteins (IAA8, IAA9, IAA12, IAA13 and IAA27), which function as repressors of early auxin response genes at low auxin concentrations by forming heterodimers with auxin response factors (ARFs) [81]; ARFs (ARF1, ARF4, ARF5, ARF6 and ARF19), which act as transcriptional repressors (e.g., IAA2, IAA3 and IAA7) by forming heterodimers with Aux/IAA proteins and promoting flowering [82,83]; and CTK response, i.e., histidine kinase 3 (AHK3), which the cytokinin-dependent flower-development-regulation pathway requires [84]; histidine-containing phosphotransfer protein 1 (AHP1), which functions as a two-component phosphorelay mediator between cytokinin sensor histidine kinases and response regulators (B-type ARRs) [85]; and two-component response regulator ARR2 (ARR2), which functions as a response regulator involved in the His-to-Asp phosphorelay signal transduction system [86]. Based on the RELs, the genes involved in promoting flowering were up-regulated during vernalization but down-regulated at freezing temperatures (Figure 8A; Appendix A). Meanwhile, the change in GA_3_ and IAA contents was also shown to be higher at vernalization compared with freezing temperatures (Figure 8B,C).

### 4.4. Genes Involved in the Circadian Clock

Flowering is often triggered when plants are exposed to appropriate day lengths, which requires the circadian clock to measure the passage of time [87]. Recent studies have found that the circadian clock is also controlled by temperatures [88]. Here, two genes were found to be involved in the circadian clock: EARLY FLOWERING 3 (ELF3), which is a transcription factor part of a circadian-clock input pathway and can regulate the initiation of flowering [89,90], and cold-regulated protein (COR27), which is a negative regulator of freezing tolerance that, together with COR28, is involved in central circadian clock regulation and in flowering promotion by binding to the chromatin of clock-associated evening genes (e.g., *ELF4*) [91,92]. According to the RELs, the genes (*ELF3* and *COR27*) favoring the initiation of flowering were up-regulated (Figure 9A; Appendix A), which accelerates the formation of leaves and eventually the physiological age during vernalization [33].

### 4.5. Genes Involved in the Downstream Floral Integrators and Meristem Identity

Flower formation occurs at the SAM and is a complex morphological event that is regulated by several genes [33]. Here, 15 genes were found to be involved in downstream floral integrators and meristem identity. Three of these genes were involved in floral integrators: FT-interacting protein 3 (FTIP3), which is required for the proliferation and differentiation of shoot stem cells in SAM [93]; FT-interacting protein 7 (FTIP7), which promotes nuclear translocation of the transcription factor homeobox 1, directly suppressing the auxin biosynthetic gene *YUCCA4* [94]; and SOC1, which integrates signals from the photoperiod, vernalization and autonomous floral induction pathways [95].

Ten genes are involved in the floral meristem identity: agamous-like MADS-box protein AGL65 (AGL65), which forms a heterodimer with the MADS-box protein AGL104 [96]; squamosa promoter-binding-like proteins (SPL1 and SPL6), which bind specifically to the consensus nucleotide sequence of the AP1 promoter [97]; APETALA 2 and -like proteins (AP2, AP2-1, AP2-2, AP2-3, AIL5, ANT and At2g41710), which are involved in initiation and development of organs, including floral organs [98,99,100,101]; AP2/ERF and B3 domain-containing transcription repressor TEM1 (TEM1), which is a transcriptional repressor of flowering time in plants that prefer long days and acts directly upon FT expression [102]; and *SOK2*, which can influence cell division orientation to coordinate cell polarization, which is fundamental for tissue morphogenesis in multicellular organisms [103,104,105]. According to the RELs, the genes involved in floral integrators and meristem identity were up-regulated during vernalization but down-regulated at freezing temperatures (Figure 9B; Appendix A). These findings further indicate that vernalization enhances the early flowering of *A. sinensis*.

### 4.6. Genes Involved in the Cold Response

Vernalization cannot be completed in the context of short exposures to cold, which might occur during cases of fluctuating temperatures, after which plants may face and adapt to low temperatures [34]. Here, 35 genes were found to be involved in the cold response, including 2 genes involved in ABA response: ABA receptors PYL3 and PYL4 (PYL3 and PYL4), which act as positive regulators of tolerance to cold stress [106]. Genes involved in the cold response also include those involved in the ETH response: ethylene-responsive transcription factors (ERF3, ERF5, ERF010, ERF011, ERF13, ERF110 and ERF118), which regulate gene expression by stress factors and by components of stress signal transduction pathways [107,108], and ETH receptor 2 (ETR2), which acts as a redundant negative regulator of ETH signaling [109]. They also include two genes involved in cold-regulated plasma membrane proteins: cold-regulated 413 plasma membrane proteins (COR413PM1 and COR413PM2), which are up- and down-regulated, respectively, in response to low temperatures [110]. In addition, they include six genes involved in cold-responsive protein kinase: cold-responsive protein kinase 1 (CRPK1), which is a negative regulator of freezing tolerance [111]; mitogen-activated protein kinases (MPK3, MPK9, MPK16 and YDA), which are associated with the ABA-activated signaling pathway in response to oxidative stress and freezing [112,113]; serine/threonine-protein kinase SRK2E (SRK2E), which is an activator of the ABA signaling pathway [114]. They also included 15 genes that were found to be involved in protein phosphatases (PP2CA, HAB1, HAB2, KAPP, PPC6-1, PPC6-7, PP2C27, PP2C38, At2g30020, Os02g0799000, At3g15260, At3g16560, At3g62260, Os06g0651600 and At4g31860), which regulate numerous ABA responses such as cold stress [115,116,117]. In addition, transcription factor TGA2.2 (TGA2.2) is involved in the defense response [118], and cell division cycle 5-like protein (CDC5) is involved in mRNA splicing and cell cycle control and may also play a role in the response to DNA damage [119,120]. According to the RELs, most of the genes involved in cold stress were up-regulated (Figure 10A; Appendix A), and the ABA contents increased in response to vernalization and freezing temperatures (Figure 10B), which trigger the stress response for the seedlings adapting to low temperatures.

### 4.7. Proposed Regulatory Networks of Flowering Genes in A. sinensis during Vernalization

Based on the functional analysis and the regulatory pathways of flowering genes in the model plant *Arabidopsis* [30,31,32], a schematic representation of the proposed regulatory networks of flowering genes in *A. sinensis* during vernalization was created and is shown in Figure 11. Briefly, during the vernalization, four pathways that regulate flowering time were observed: the vernalization pathway, the autonomic pathway, the age pathway and the GA (hormone) pathway. In the vernalization pathway, the expression of *FLC* is inhibited by the genes *VIL1*, *VIL2*, *VRN1-L*, *LHP1* and *MAF1* but activated by the genes encoding the PAF1 complex (e.g., VIP4, ATX2 and ATX), the SWR1 complex (ARR6 and PIE1) and FRIGIDA-like proteins (FRL3, FES and SUF4) and regulated by proteins of the polycomb group (CLF and FIE2). In the autonomic pathway, the expression of *FLC* is inhibited by the genes involved in RNA binding (*FCA*, *FPA* and *FLK*), the RNA process (*FY*) and chromatin modification (*FLD* and *LD*). In the age pathway, the expression of *SOC1* is promoted by the genes encoding SPLs (SPL1 and SPL6), which can be positively regulated by the genes involved in sucrose metabolism (e.g., *TPS1*, *SUS2* and *SPS*). In the GA (hormone) pathway, the expression of *SOC1* is promoted by the genes involved in the CTK response (*AHK3*, *AHP1* and *ARR2*) but inhibited by the genes involved in ETH response (*RAP2-3* and *RAP2-7*) and IAA (e.g., *GH3.6*, *LAX2* and *IAA8*); the expression of *AP1* is promoted by the DELLA protein, which is inhibited by the genes involved in the GA response (*GID1B*). Additionally, the expression of *FT* (*FTIP3* and *FTIP7*) is promoted by the genes involved in the circadian clock (*ELF3* and *COR27*), as well as AP2 and -like proteins (e.g., AP2, AP2-1 and AIL5), but inhibited by the gene *TEM1*. The expression of *SOC1* is also promoted by the gene *AGL65*. The coordinated expression of these genes during vernalization confers the transition of seedlings from vegetative growth to flowering of *A. sinensis*.

## 5. Conclusions

The DEGs observed in *A. sinensis* during vernalization strongly suggest that transcription-based regulation occurs for the transition of seedlings from vegetative growth to flowering. The expression level of genes involved in flowering and the cold response during vernalization were almost consistent with changes in sugars and hormone contents. There are four pathways of genes that are required for regulating flowering, including the vernalization pathway, the autonomic pathway, the age pathway and the GA (hormone) pathway. While genes involved in flowering during vernalization have been mapped here, additional studies are required to determine the causative role of these genes in promoting or inhibiting the expression of the central genes (e.g., *VINs*, *SOC1* and *FT*).

## Figures and Tables

**Figure 1 plants-11-01355-f001:**
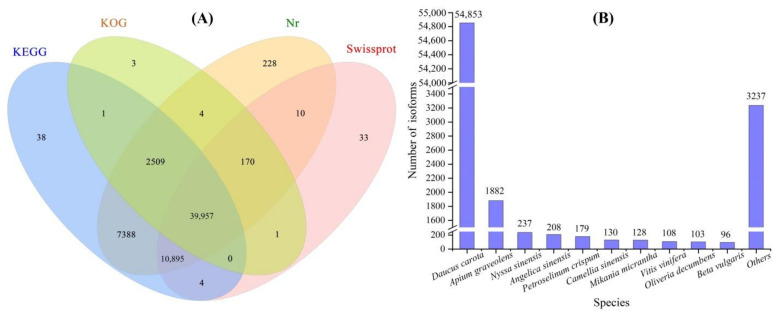
Basic annotation of full-length isoforms against KEGG, KOG, Nr and Swiss-Prot databases (**A**) and top 10 species in terms of distribution of the isoforms against Nr (**B**).

**Figure 2 plants-11-01355-f002:**
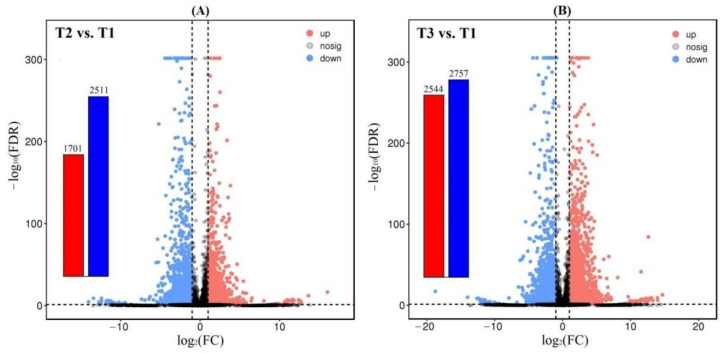
Volcano plot of differential expression at T2 vs. T1 (**A**) and T3 vs. T1 (**B**).

**Figure 3 plants-11-01355-f003:**
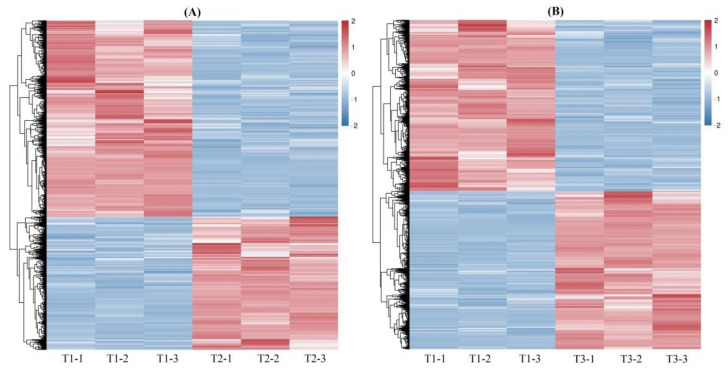
Cluster heat map of the DEGs at T2 vs. T1 (**A**) and T3 vs. T1 (**B**).

**Figure 4 plants-11-01355-f004:**
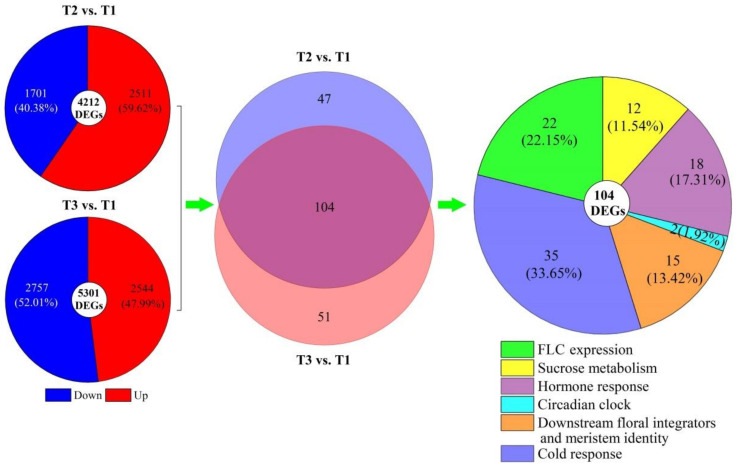
Distribution and classification of DEGs in *A. sinensis* at T2 vs. T1 and T3 vs. T1.

**Figure 5 plants-11-01355-f005:**
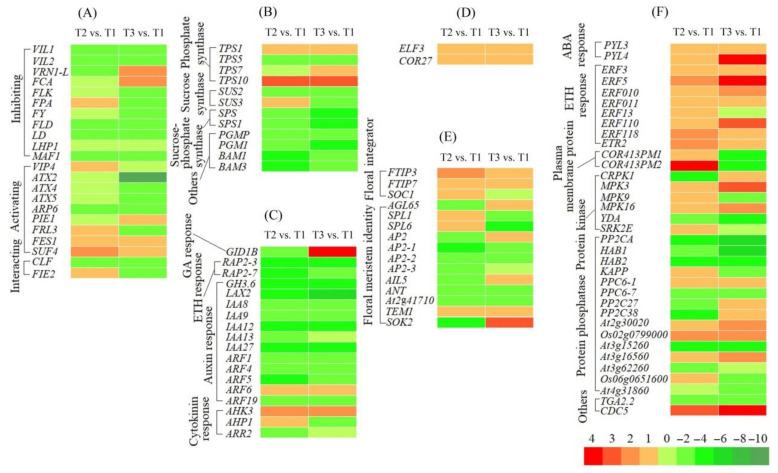
Heat map of the 104 DEGs involved in flowering at T2 vs. T1 and T3 vs. T1. The differential expression level is based on FPKM values; gene abbreviations are provided in Appendix A. The subfigures (**A**): FLC expression, (**B**): sucrose metabolism, (**C**): hormone response, (**D**): circadian clock, (**E**): downstream floral integrators and meristem identity, and (**F**): cold response.

**Figure 6 plants-11-01355-f006:**
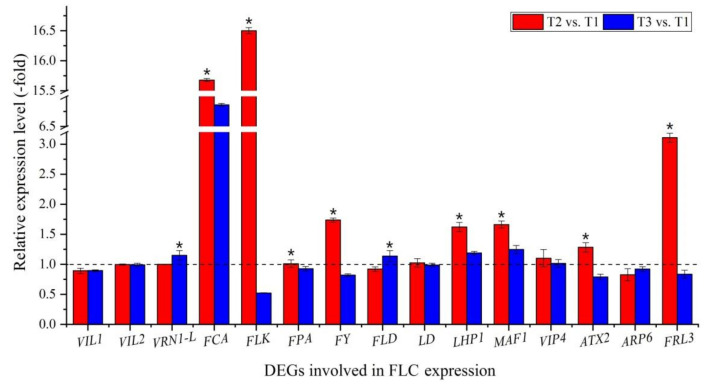
Expression levels of 15 genes involved in FLC expression at T2 vs. T1 and T3 vs. T1. Here and below, “*” represents a significant difference at the *p* < 0.05 level between T2 vs. T1 and T3 vs. T1 for the same gene.

**Figure 7 plants-11-01355-f007:**
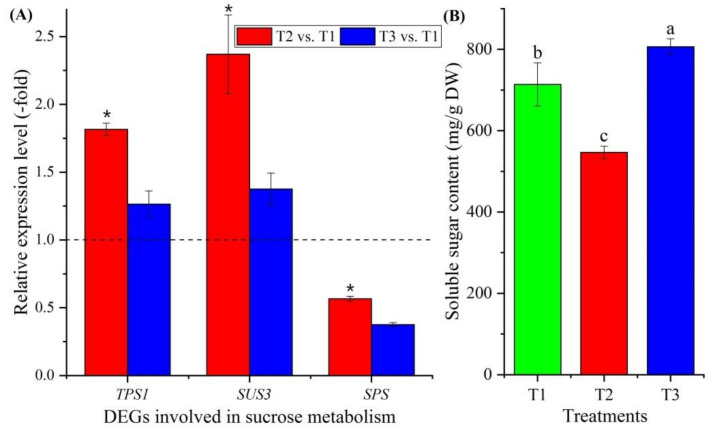
Expression levels of 3 genes involved in sucrose metabolism at T2 vs. T1 and T3 vs. T1 (**A**), and changes in soluble sugars contents at T1, T2 and T3 (**B**). Here and below, “*” represents a significant difference at the *p* < 0.05 level between T2 vs. T1 and T3 vs. T1 for the same gene; the use of different letters represents a significant difference (*p* < 0.05) at different temperature treatments.

**Figure 8 plants-11-01355-f008:**
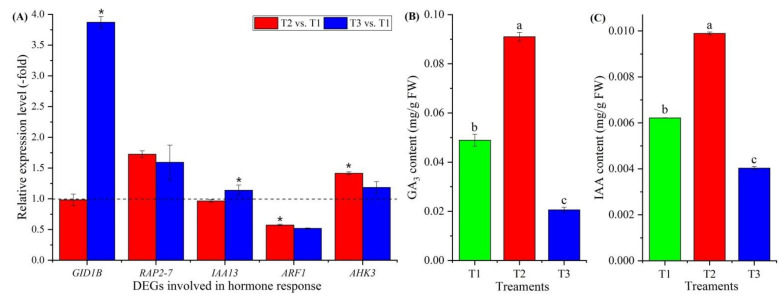
Expression levels of 5 genes involved in hormone response at T2 vs. T1 and T3 vs. T1 (**A**), and changes in GA_3_ and IAA contents at T1, T2 and T3 (**B**,**C**). Here and below, “*” represents a significant difference at the *p* < 0.05 level between T2 vs. T1 and T3 vs. T1 for the same gene; the use of different letters represents a significant difference (*p* < 0.05) at different temperature treatments.

**Figure 9 plants-11-01355-f009:**
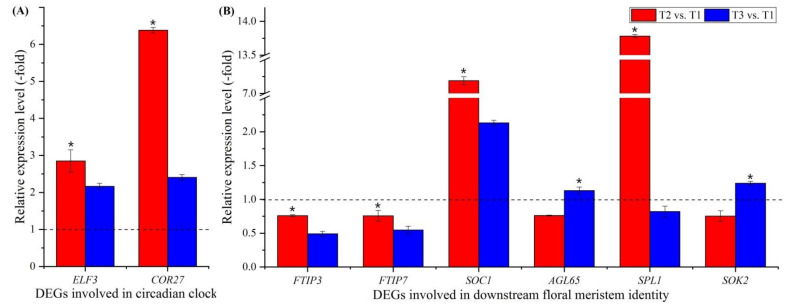
Expression levels of 2 genes involved in the circadian clock (**A**), and 6 genes involved in downstream floral meristem identity at T2 vs. T1 and T3 vs. T1 (**B**). Here and below, “*” represents a significant difference at the *p* < 0.05 level between T2 vs. T1 and T3 vs. T1 for the same gene; the use of different letters represents a significant difference (*p* < 0.05) at different temperature treatments.

**Figure 10 plants-11-01355-f010:**
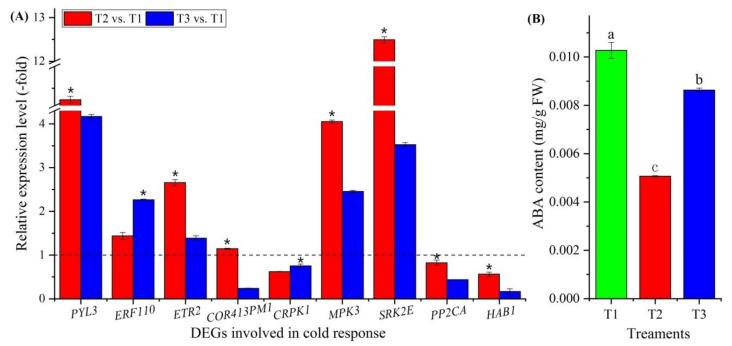
Expression levels of 9 genes involved in cold response (**A**) and changes in ABA contents at T1, T2 and T3 (**B**). Here and below, “*” represents a significant difference at the *p* < 0.05 level between T2 vs. T1 and T3 vs. T1 for the same gene; the use of different letters represents a significant difference (*p* < 0.05) at different temperature treatments.

**Figure 11 plants-11-01355-f011:**
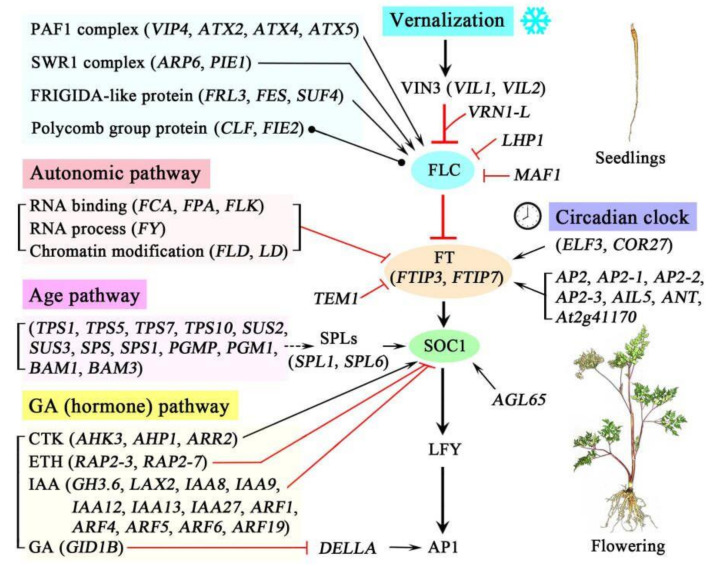
Schematic representation of the proposed regulatory networks of flowering genes in *A. sinensis* during vernalization. The arrows indicate a promotion, with T-ends with red color indicating an inhibition, round dots at both ends mark an interaction without a known direction, and dashed lines indicate an indirect interaction.

**Table 1 plants-11-01355-t001:** Summary of Illumina sequencing data of *A. sinensis* for T1, T2 and T3.

	T1	T2	T3
**Raw data**			
Data of reads number (million)	9.14	9.19	8.63
Q20 (%)	97.87	97.87	97.99
Q30 (%)	93.88	93.87	94.20
GC (%)	42.90	42.99	43.07
**Filtered data**			
Data of reads number (million)	9.12	9.16	8.61
Q20 (%)	97.97	97.97	98.09
Q30 (%)	94.01	94.10	94.34
GC (%)	42.85	42.94	43.01
**Mapped data against full-length isoform**			
Data of unique mapped reads (million)	5.33	5.50	5.23
Data of multiple mapped reads (million)	2.27	2.31	2.05
Exon rate (%)	100	100	100

Note: T1 (0 °C 14 d), uncompleted vernalization; T2 (0 °C 60 d), completed vernalization; T3 (−3 °C 125 d), avoided vernalization.

## Data Availability

The datasets are available at NCBI, with BioSample accession SAMN24046640 to SAMN24046648; SRA accession SRR17235563 to SRR17235571 (9 objects) with every treatment three replicates; and the BioProject’s metadata are available at https://www.ncbi.nlm.nih.gov/bioproject/PRJNA789039, accessed on 14 December 2021.

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
