# Peer review of "Regulatory Networks of Flowering Genes in *Angelica sinensis* during Vernalization"

_plants, 2022, doi:10.3390/plants11101355_

Round 1

Reviewer 1 Report

Authors have addressed my concerns sufficiently. However, authors should provide sequence details of the isoforms as supplementary material.

Author Response

Many thanks for your and reviewer’s comments that are helpful to improve our paper much better now. We have tried to address and correct each comment. Attachments below with our responses are shown in bold. Revised parts are highlighted in red in the manuscript.

According to your comments, the sequence details of the isoforms involved in the 104 co-expressed genes at T2 vs. T1 and T3 vs. T1 have been provided as Table S2 in the Supplementary material. Meanwhile, the description has been added in the text: “The sequence details of the isoforms involved in the 104 co-expressed genes are shown in Table S2.” (Page 6, lines 225-226)

Reviewer 2 Report

This review concerns the re-submission of a manuscript by Luo et al. describing a study of the mecanisms regulating flowering of A. sinensis upon vernalization. The study is mainly based on transcriptomic approaches, first by RNAseq to screen for target genes of interest followed by RT-qPCR for confirmation. It includes also phytohormones and sugar dosage.

For re-submission, the authors did some changes in the text and included new tables in the supplementary data. Considering my first review of the original manuscript, these modifications don't change my views.

I will just insist on two points:

1/ Poor interpretation/discussion of the results. I know it's a difficult point that could be source of misunderstanding and that is, after all, based on my own vision of what should be a proper scientific discussion based on facts and bibliography. Consequently, I will not insist on this point anymore and will let the editorial board take their own decision.

2/ However, I still have a problem, which is a technical point, on the way some data were obtained. It concerns the use of "actin" (again, which one? How does it behave when compared to other actin genes? Or is there only one actin gene in A. sinensis genome?...I don't know!) as a reference gene in the RT-qPCR validations. Well, the answer of the authors to this point was just to mention papers they already published using the same gene. Well, sorry, but this is definitely not enough. Indeed, the use use of an inproper reference gene in such experiments could be of huge influence on the results. To try to stress that out, I spent some time comparing RNAseq (figure 5 and FPKM values of the tables provided in supplementary data) and RT-qPCR (Fig. 6 to 10) data. I know that RNAseq data are less precise but the authors claim that they are confirmed (for "almost all of them") by the RT-qPCR data. Well, I don't agree with that. Indeed, the tendancies (up-down when compared to T1 conditions) could be intrepreted as being the same. However, the tresholds are, sometimes, considerably different. And maybe it's the case in reality.....but I am not sure because of the use of the "actin" gene as a reference. If you don't prove experimentaly that, in your conditions, "actin" gene expression is unchanged...well, you can't be sure too. The only way is to use multiple reference genes and to use the best one, As it is mentionned in the paper of Willems et al., ref 27 of the manuscript. Finally, these treshold differences are never discussed by the authors for alternative interpretation. This is connected to my point N°1.

That's the main reason why I consider that the manuscript can't be accepted without answering experimentaly to my second point.

Author Response

Many thanks for your and reviewer’s comments that are helpful to improve our paper much better now. We have tried to address and correct each comment. Attachments below with our responses are shown in bold. Revised parts are highlighted in red in the manuscript.

1>Thanks very much for your kind review. We have tried our best to discuss the results by citing previously published articles, firstly describing the functions of candidate genes, and finally proposing regulatory networks of flowering genes in A. sinensis during vernalization based on the genes functions in this study and the model plant Arabidopsis. We hope you and the editor office could understand the writing format to show the findings of our studies.

2>Indeed, the actin gene was used as an only one to evaluate the expression level of 40 candidate genes in this study. In order to obtain the precise estimation, the standard curve of actin gene along with the R2 value and PCR efficiency has been performed in this study, which has been provided as Figure S3 and Figure S4 in the Supplementary material.

Actually, based on the three biological along with three technical replications for the 40 candidate genes, both the lower standard deviations and significant differences between different treatments (Figures 6 to 10) also demonstrate that the actin can be used as a reference gene.

In order to show the stable expression and precise results of the expression level for 40 candidate genes, the information has been added in the text:

1) Herein, the cycle threshold (Ct) values and standard curves of the ACT gene at different volumes (0.25, 0.5, 1.0, 1.5, 2.0 and 3.0 μL) was built to correct the gene expression level (Figure S3 and Figure S4) (Page 3, lines 142-145)

2) In order to obtain the precise estimation of PCR efficiency, each experiment for qRT-PCR validation was performed with three biological replicates, along with three technical replicates. (Page 4, lines 152-154)

3) The significant differences of the 40 candidate genes from Figures 7 to 10 have been analyzed: “The “*” represents a significant difference at p < 0.05 level between T2 vs. T1 and T3 vs. T1 for the same gene”. (Page 8, lines 248-250)

Additionally, it would be nice to select more reference genes to evaluate the gene expressive level via qRT-PCR, based on our isoforms obtained by RNA-seq. According to your suggestion, these works will be further performed in our next experiments. Currently, the pandemic of Covid-19 has been delaying our research process and restricting our researchers to the Lab for a long time. We hope that the all you reviewers and editor office can understand the current manuscript without selecting the best reference gene from the multiple reference genes.

This manuscript is a resubmission of an earlier submission. The following is a list of the peer review reports and author responses from that submission.

Round 1

Reviewer 1 Report

This paper tries to investigate regulatory networks of flowering of A. sinensis during vernalization. The authors used RNA sequencing to monitor comprehensive transcriptomic behaviors. Identified DEGs were validated by qRT-PCR. They showed 4 pathways as candidate regulatory circuits. As a whole, this study may be technically sound.  I had only conmments about the method descriptions.

  • The description of reference genome is lacking. The assembly version and the citation are missing.
  • The method used is ambiguous. For example, did you use both EdgeR and DESeq2 to identify DEGs? Did you integrate their results from those?

Reviewer 2 Report

In the manuscript titled "Regulatory Networks of Flowering Genes in Angelica sinensis during Vernalization" by Luo et al., authors profiled gene expression at three flowering time points. The work is well performed and well presented. However, there is one main concern with the current version which I have listed below.

In figure 1 authors show the comparison of isoforms compiled from this study with various database in species. There they show that only 208 transcripts map to Angelica sinensis. What lead to such low count from the targetted species? 

Authors should also compare the isoforms with results from two previous studies.

 Gao et al., Scientific Reports 2021 (https://www.nature.com/articles/s41598-021-92494-4#Sec20)

Yu et al., Scientific Reports 2019 (https://www.nature.com/articles/s41598-019-46414-2)

Reviewer 3 Report

The manuscript of Luo et al. aims to decipher the regulatory networks involved in the control of flowering by vernalization in Angelica sinensis.

The experimental plan is only based on identification of differentially expressed genes between contrasted temperature growth conditions that were shown to activate or inhibit flowering in reponse to cold. The methods used are sufficiently described and are mainly based on RNAseq based approaches. The expression of interesting genes identified through RNAseq data analysis was confirmed (or not) using RT-qPCR. I should mention that the autors should pay attention to the use of proper RT-qPCR reference genes. I am not confinced, reading the manuscript, that this is the case. The autors only describe the use of the "actin" gene as reference for RT-qPCR. Which one? Was it shown, including in previously published work, that this "actin" gene has a stable expression in the growth conditions used in Angelica sinensis? These informations are missing and are primordial to obtain solid RT-qPCR data.

My main concern about this manuscript is its lack of consistancy in terms of interpretation of the data. The discussion is really basic (description of a list of genes) and is redondant with the results data description. Each chapter of the conclusions is ending with a single sentence  as "in this study,...". This is definitely not enough as a discussion of the data. The final chapter of the discussion, including the final figure, is a copy/paste of what was already shown since a long time in other plant species. Maybe am I wrong to claim that, as I am not a specialist in the field of flowering, but the lack off originality is striking.

To continue, as a non specialist on flowering induction pathways, I anyway have a big problem with a specific point: the autors, according to what is written (for example in figure 11...but also in the text) seem to claim that FLC and SOK2 genes are the same gene. Well, I am afraid it is not the case. And, SOK2 is definitively not involved in the flowering pathway. I could be wrong on this point but the autors should make this clear in their manuscript.

Another big problem for me is that they don't find, as differentially expressed, the FLC gene in their data despite the fact it's one of the major genes differentially expressed upon vernalization. Or is it SOK2 which is differentially expressed? SOK2 gene is next to FLC gene in Arabidopsis but they are different genes. And is it the same in Angelica sinensis?

Finaly, I found in the manuscript bibliography a reference (n°12) which is a self-citation (which is not a big deal) but unpublished ("under review"). Well, this must not be allowed in a proper scientific manuscript.

I forgot to add that English could be improved, but a s a minor point.

To conclude, considering my analysis of the manuscript (which is just my opinion), I do recommand the rejection of this manuscript for publication in Plants. I encourage the autors to re-writte it to improve the intermretation of the data, which could be interesting to the community. However, additional experimental work is necessary.